# Postharvest Preservation of the New Hybrid Seedless Grape, 'BRS Isis', Grown Under the Double-Cropping a Year System in a Subtropical Area

**Saeed Ahmed [1], Sergio Ruffo Roberto [1,*], Khamis Youssef [2], Ronan Carlos Colombo [1], Muhammad Shahab [1], Osmar José Chaves Junior [1], Ciro Hideki Sumida [1] and Reginaldo Teodoro de Souza [3]**

[1] Agricultural Research Center, Londrina State University, Celso Garcia Cid Road, km 380, P.O. Box 10.011, 86057-970 Londrina, Brazil
[2] Agricultural Research Center, Plant Pathology Research Institute, 9 Gamaa St., Giza 12619, Egypt
[3] Embrapa Grape and Wine, 515 Livramento Drive, 95700-000 Bento Gonçalves, Brazil
[*] Correspondence: sroberto@uel.br; Tel.: +554-333-714-774

**Abstract:** 'BRS Isis' is a new hybrid seedless table grape tolerant to downy mildew with a good adaptation to the tropical and subtropical climates. Gray mold, caused by *Botrytis cinerea* Pers. ex Fr. is known as the most important postharvest mold in table grapes, causing extensive losses worldwide. As the postharvest behavior of 'BRS Isis' is still unknown, the objective of this work was to evaluate the postharvest preservation and *B. cinerea* mold control of this new grape cultivar, grown under the double-cropping a year system. Grape bunches were purchased from a field of 'BRS Isis' seedless table grapes trained on overhead trellises located at Marialva, state of Parana (South Brazil). Grapes were subjected to the following treatments in a cold room at $1 \pm 1$ °C: (i) Control; (ii) $SO_2$-generating pad; (iii) control with bunches inoculated with the pathogen suspension; (iv) $SO_2$-generating pad with bunches inoculated with the pathogen suspension. The completely randomized experimental design was used with four treatments, each including five replicates. The incidence of gray mold and other physicochemical variables, including bunch mass loss, shattered berries, skin color index, soluble solids (SS), titratable acidity (TA), and SS/TA ratio of grapes, were evaluated at 50 days after the beginning of cold storage and at seven days at room temperature ($22 \pm 2$ °C). The 'BRS Isis' seedless grape, packaged with $SO_2$-generating pads and plastic liners, has a high potential to be preserved for long periods under cold storage, at least for 50 days, keeping very low natural incidence of gray mold, mass loss, and shattered berries.

**Keywords:** *Vitis vinifera* (L.); $SO_2$ pads; *B. cinerea* mold; grape quality

## 1. Introduction

'BRS Isis' is a new hybrid seedless table grape obtained by the crossing of CNPUV 681–29 (Arkansas 1976 × CNPUV 147-3 ('Niagara White' × 'Venus')) × 'BRS Linda'. This cultivar was released in 2013 and is tolerant to downy mildew *Plasmopara viticola* (Berk. & M.A. Curtis) Berl. & De Toni, the main vine disease in subtropical humid areas. It presents high bud fertility with 2–3 great inflorescences per shoot, with a natural weight of 375 g, and without the use of growth regulators, making it a high yielding grape. The bunch is medium-sized and predominantly cylindrical-winged, while the berry is medium-sized, reddish, elliptical, and firm and has colorless flesh and neutral flavor with traces of large, fleshy rudimentary seeds [1]. This new seedless and early season cultivar has the ability to gain the attention of consumers from domestic and international markets as there has been a significant demand of table grape supply for extended periods throughout the year worldwide.

Grapes are non-climacteric fruits with a relatively low physiological activity and are subject to serious postharvest problems during cold storage, such as mold, mass loss stem browning, shattered berries, wilting, and shriveling of berries. Thus, these factors are the main barriers for long-term storage of table grapes [2–4].

*Botrytis cinerea* Pers. Fr., is known to be the most important postharvest pathogen causing gray mold of table grapes [5,6]. Infection caused by this fungus remains inactive in the field unless it gets favorable environmental conditions, i.e., fruit injuries that assist pathogen development [7,8]. Even a small infection on a single berry can damage the whole lot of grapes, and if it is not noticed at pre-harvest stage, during packaging, or during shipment, it may progress and spread the infection in postharvest or during the cold storage period of table grapes, even at low temperatures [9–11].

Cold storage, where only temperature and relative humidity are controlled in the chamber, is one of the main methods to maintain the fruit quality. Thus, the reduction of temperature, up to a certain limit, increases the quality preservation and extends the period of fruit supply to the consumer market [12]. After harvesting, bunches are pre-cooled as soon as possible to remove field heat and reduce water loss [11,13]. For extended export and shipment purposes, the cold storage temperature must be kept optimum and constant because any disturbance can initiate the growth of fungi, mainly *B. cinerea* [14].

The postharvest control of this pathogen is difficult, as most countries no longer allow the application of synthetic fungicides on bunches. Combined with cold storage, different pre- and postharvest techniques can be used to control gray mold, such as the use of sulfur dioxide ($SO_2$) generating pads, which is the most common method worldwide [15–17]. The slow release $SO_2$-generating pads contain sodium metabisulfite ($Na_2S_2O_5$) as am active ingredient enclosed in a sheet of plastic and paper, which used in packing materials by releasing a low and continual dose of $SO_2$ with contact to humidity to eliminate/reduce *B. cinerea* spores.

The $SO_2$-generating pads are highly effective in controlling and killing the spores of *B. cinerea*, but also can result in unwanted situations, such as bleaching and shattered berries. Other studies have also shown that grape hairline splits, commonly associated with significant water loss, are also induced by excessive $SO_2$ doses. However, high levels of $SO_2$ can also result in fruit damage, unpleasant aftertaste, and allergies. Based on these findings, it is recommended to use a minimal dose of $SO_2$ that allows adequate protection from mold without reducing the berry quality in order to avoid these situations [18,19].

As there is a lack of information regarding the cold storage of the 'BRS Isis' seedless grape, it is very important to know the behavior of this new hybrid cultivar grown under the double-cropping a year system, especially for long-distance and international markets. Under this system, two crops per year are achieved (summer and off-season crops). Summer crops start from the end of grapevine dormancy in late winter and harvest is obtained in summer, while, for off-season crops, vines are pruned after summer crops and forced to sprout once more using budburst stimulators, and harvest occurs through autumn. The core difference between both crops is that in the summer crop, the rate of some fungal infection is quite low, while on the other hand, in an off-season crop, the incidence of fungus diseases is high because of favorable environmental conditions that promote the infection and can restrain long-distance transportation of table grapes [20,21].

The objective of this work was to evaluate the postharvest preservation and control gray mold of the 'BRS Isis' seedless grape grown under the double-cropping a year system in subtropical conditions.

## 2. Materials and Methods

### 2.1. Experimental Location

Table grapes were purchased from a field of 'BRS Isis' seedless grapes, grafted on 'IAC 766' rootstock from 2-year-old vines, trained on overhead trellises located at Marialva, state of Parana (PR) (South Brazil) (23°29 S, 51°47 W, elevation 570 m), with a history of gray mold. The vines were grown

under the double-cropping a year system, and the fruit samples were collected from two consecutive crop seasons.

## 2.2. Treatments and Storage

Grapes were harvested at full ripeness when the content of the berry soluble solids reached around 14 °Brix [21,22]. Bunches were selected free from any disorders and standardized according to bunch shape, size, and mass, and subjected to the following treatments into a cold chamber at 1 ± 1°C: (i) Control; (ii) SO$_2$-generating pad; (iii) control with bunches inoculated with *B. cinerea* suspension; (iv) SO$_2$-generating pad with bunches inoculated with *B. cinerea* suspension. The slow release SO$_2$-generating pad used in treatments (ii) and (iv) (Osku Hellas®, Grapeguard, Santiago, Chile) contain 73.5% of the active ingredient (Na$_2$S$_2$O$_5$), with 26 cm × 36 cm of dimensions.

A fungal suspension was prepared, according to the standard protocol, using a *B. cinerea* isolate (BCUEL-1), isolated from infected grapes with representative symptoms of the disease, according to Youssef and Roberto [23]. The suspensions were diluted with sterilized distilled water to get a final concentration of 10$^6$ conidia mL$^{-1}$ using a hemocytometer with 1/10 mm deep (Neubauer Boeco, Hamburg, Germany). As the incidence of gray mold can be low, depending on the season, the grapes from treatments (iii) and (iv) were inoculated with a pathogen suspension, according to Youssef and Roberto [23]. A volume of 200 mL of inoculums was sprayed on each 50 kg of grapes until dripping, using a plastic sprayer. The control consists of bunches treated only with distilled water. All bunches were air dried at room temperature (RT) before packaging.

The grapes of all treatments were packaged as follows: A micro-perforated plastic liner (1% of the ventilated area, Suragra S.A., San Bernardo, Chile) was placed inside carton boxes; grapes were placed inside the box; an SO$_2$-releasing pad was placed on top only for treatments (ii) and (iv); and the liner was sealed. The SO$_2$-releasing pad fully covered the grapes.

The boxes were placed in a cold room storage at 1 ± 1 °C and at high relative humidity (>95%). As 'BRS Isis' is a new cultivar and there is no information available regarding its cold storage performance, after 30 days of cold storage, the boxes were opened for inspection, and as the bunches of all treatments were intact, with fresh and green stems, free of any mold or injuries, it was decided to keep the boxes in the chamber for an extended period, i.e., 50 days, followed by 7 days of shelf-life at RT (22 ± 2 °C). The completely randomized experimental design was used as a statistical model with four treatments and five replications, and each plot consisted of one carton box (each measuring 23 cm × 16 cm × 9 cm (4 kg capacity)).

## 2.3. Evaluation of Gray Mold Incidence

The incidence of gray mold on grapes was evaluated at 50 days after the beginning of cold storage and at 7 days at 22 ± 2 °C after the end of cold storage. The disease incidence was then calculated: Disease incidence (% of diseased berries) = (number of infected berries/total number of berries) × 100 [23].

## 2.4. Physicochemical Analysis

The grape physicochemical analysis was evaluated twice: (i) 50 days after the beginning of cold storage; (ii) at seven days at 22 ± 2 °C following the cold storage period, using 10 berries for each box (replication). The bunch mass loss as a percentage was calculated as follows: Mass loss (%) = ((mi−ms)/mi) × 100, where mi is the initial mass and ms is the mass at the examined time [24]. Shattered berries were evaluated by calculating the separated grape berries from the bunch stem and were expressed as a percentage of the total number of berries: Shattered berries (% of diseased berries) = (number of shattered berries/total number of berries) × 100.

The berry color was investigated using a colorimeter CR-10 (Konica Minolta®, Tokyo, Japan) to get the following variables from the equatorial portion of grape berries (*n* = 2 per berry): *L** (lightness), *C** (chroma) and *h°* (hue angle). The color index for red grapes (CIRG) was then calculated using the formula CIRG = (180 − *h°*)/(*L** + *C**) [25]. Ten berries were collected from each replicate to be

investigated. Lightness rates range from 0 (black) to 100 (white). Chroma indicates the purity or intensity of color and the distance from gray (achromatic) toward a pure chromatic color and is measured from the *a** and *b** values of the CIELab scale system, starting from zero for a completely neutral color, and does not have an arbitrary end, but the intensity increases with magnitude. Hue refers to the color wheel and is calculated in angles; green, yellow, and red correspond to 180, 90, and 0°, respectively [26–28].

For the chemical analysis, 10 berries were collected from each replicate. To determine soluble solid (SS) content and titratable acidity (TA), samples were crushed, and the juice was used. For SS, some juice drops were analyzed using a digital refractometer (Krüss DR301-95; A. Krüss Optronic, Hamburg, Germany) with automatic temperature compensation at $20 \pm 1$ °C, and the results were presented as °Brix. TA was determined using a dropwise titration with 0.1 N NaOH using 10 mL of grape juice diluted in 40 mL of distilled $H_2O$, and pH = 8.2 was considered as the endpoint. The results were presented as tartaric acid (%) [29]. The SS/TA ratio was used to express the maturation index of grape berries.

### 2.5. Statistical Analysis

All data were subjected to an analysis of variance (ANOVA) by using Sisvar® software (UFLA, Lavras, Brazil). The mean values of treatments were compared by using Fisher's protected least significant difference (LSD) test and judged at $p \leq 0.05$ levels. Percentage data were arcsine transformed to normalized variance. Data in the tables or charts are the untransformed percentage of rotted grape berries.

## 3. Results

### 3.1. Incidence of Gray Mold

The disease incidence found at 50 days of cold storage was considered low in both seasons, and no significant differences were observed when grapes were subjected to control and $SO_2$-generating pad treatments only (Figures 1 and 2). On the other hand, when grapes were inoculated with *B. cinerea* suspension, the $SO_2$-generating pads significantly decreased the incidence of gray mold of grapes harvested in the summer crop season, as compared to the control with bunches inoculated with *Botrytis*. In the case of the off-season crop, although the incidence of gray mold was higher in grapes of the control and $SO_2$-generating pads, both inoculated with *B. cinerea* suspension, no significant differences were observed between them.

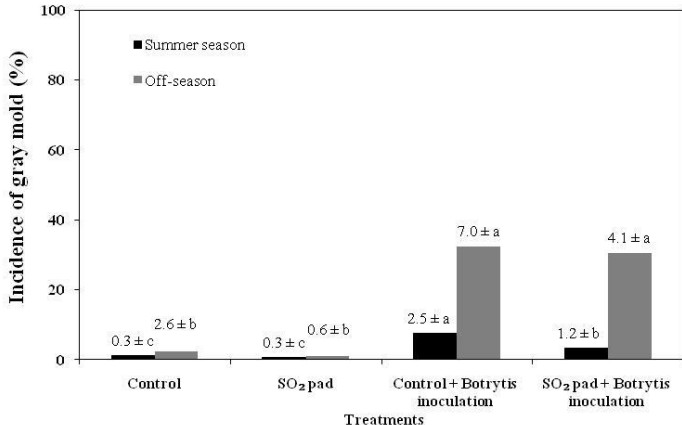

**Figure 1.** Incidence of gray mold (% of diseased berries) at 50 days of cold storage of 'BRS Isis' seedless table grapes during summer and off-season crops. Columns followed by different letters, in relation to the treatments within each individual crop, are statistically different, according to Fisher's protected LSD test ($p \leq 0.05$).

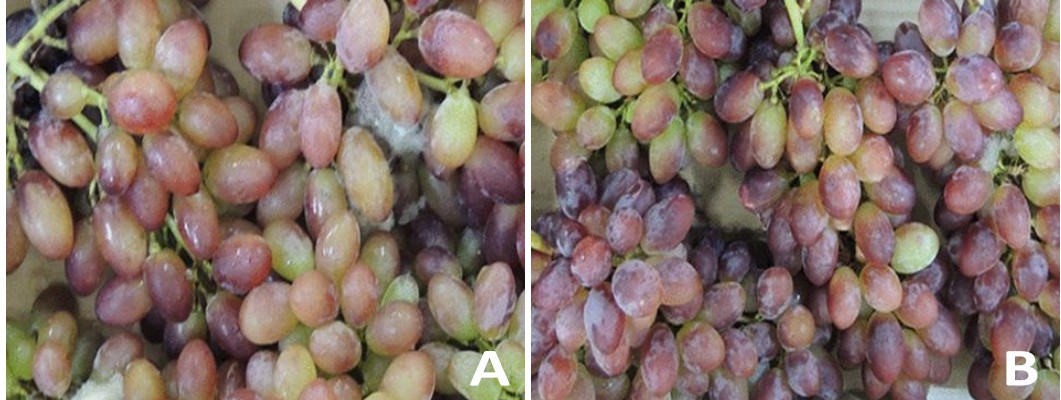

**Figure 2.** Bunches of 'BRS Isis' seedless table grapes at 50 days under cold storage. (**A**) Control; (**B**) SO$_2$-generating pads.

It was observed that the incidence of gray mold was higher in the off-season crop (~30%) when grapes were inoculated with *B. cinerea* suspension. This situation was also found after the 7 day period at 22 ± 2 °C, where the gray mold incidence (~50%) was higher as compared to the 50 day period of cold storage (Figure 3).

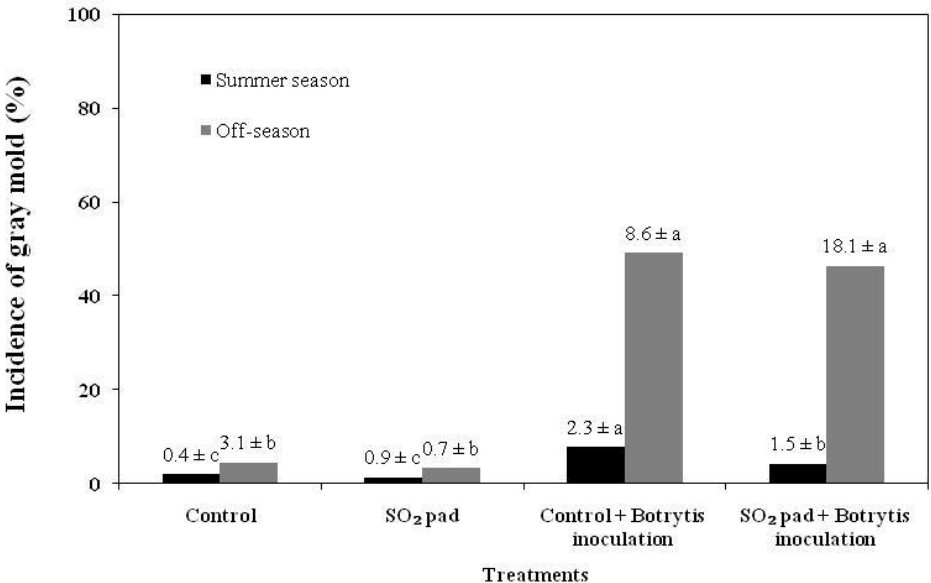

**Figure 3.** Incidence of gray mold (% of diseased berries) at seven days at 22 ± 2 °C of 'BRS Isis' seedless table grapes during summer and off-season crops. Columns followed by different letters, in relation to the treatments within each individual crop, are statistically different, according to Fisher's protected LSD test ($p \leq 0.05$).

After seven days of 22 ± 2 °C, no significant differences were found among treatments with SO$_2$-generating pads in comparison to the control (with no *B. cinerea* inoculation) in both crop seasons. However, when grapes were inoculated with *B. cinerea* suspension, significant differences were observed in the summer season crop, where the SO$_2$-generating pads resulted in lower gray mold incidence (4.2%) in comparison to the control with inoculated grapes (7.8%), while in the case of the off-season crop, no differences were found (Figures 3 and 4).

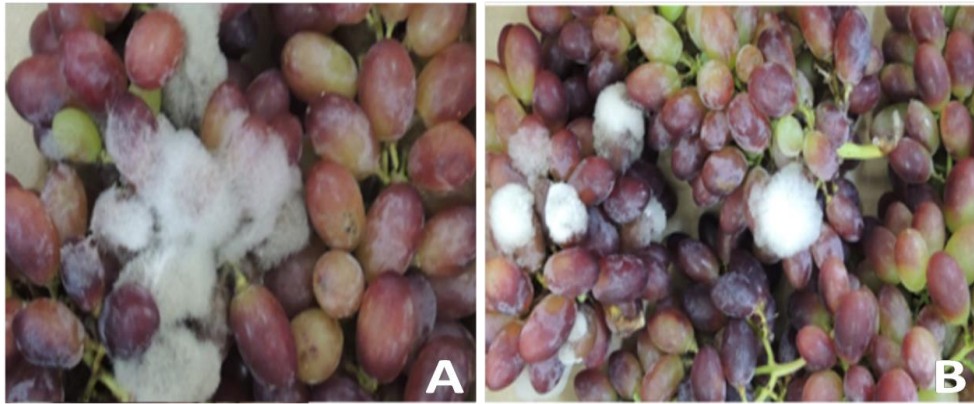

**Figure 4.** Bunches of 'BRS Isis' seedless table grapes at seven days at 22 ± 2 °C; (**A**) control with *B. cinerea* inoculation; (**B**) SO$_2$-generating pads with *B. cinerea* inoculation.

### 3.2. Physical Characteristics of Grapes

There were no significant differences among treatments for mass loss at 50 days of cold storage in the summer crop, and means varied from 5.4 to 7.0%, while in the case of the off-season crop, significant differences were noted, where both *B. cinerea* inoculated treatments (the control and SO$_2$-generating pad) showed higher mass loss as compared to non-inoculated treatments (Table 1).

**Table 1.** Mass loss (%), shattered berries (%), and color index for red grapes (CIRG) of the 'BRS Isis' seedless table grape at 50 days of cold storage and at seven days at 22 ± 2 °C during summer and off-season crops.

| Treatments | Mass loss (%) | | Shattered berries (%) | | CIRG | |
|---|---|---|---|---|---|---|
| | At 50 days of cold storage | | | | | |
| | Summer season | Off-season | Summer season | Off-season | Summer season | Off-season |
| Control | 6.3 ± 2.4 | 3.1 ± 0.4 b | 0.0 ± 0.0 | 1.2 ± 2.5 | 3.8 ± 0.4 | 5.1 ± 0.3 |
| SO$_2$ pad | 5.6 ± 1.4 | 3.0 ± 0.2 b | 0.3 ± 0.3 | 0.7 ± 0.5 | 4.4 ± 0.2 | 5.4 ± 0.3 |
| Control + Bot | 5.4 ± 0.9 | 3.7 ± 0.3 a | 0.7 ± 0.6 | 3.1 ± 1.0 | 4.5 ± 0.5 | 5.4 ± 0.4 |
| SO$_2$ pad + Bot | 7.0 ± 2.6 | 3.6 ± 0.3 a | 0.2 ± 0.4 | 2.1 ± 1.1 | 4.6 ± 0.5 | 5.5 ± 0.2 |
| F value | 0.7 [NS] | 5.4 * | 2.5 [NS] | 2.6 [NS] | 3.4 [NS] | 1.5 [NS] |
| | At seven days at 22 ± 2 °C | | | | | |
| | Summer season | Off-season | Summer season | Off-season | Summer season | Off-season |
| Control | 5.8 ± 0.3c | 1.2 ± 0.1c | 0.1 ± 0.3b | 1.8 ± 2.2b | 4.0 ± 0.6 | 4.8 ± 0.4 |
| SO$_2$ pad | 7.2 ± 0.8ab | 1.6 ± 0.4ab | 0.7 ± 0.6b | 1.5 ± 1.3b | 3.8 ± 0.3 | 4.6 ± 0.3 |
| Control + Bot | 7.6 ± 0.5a | 1.3 ± 0.2bc | 1.5 ± 0.5a | 12.1 ± 9.3a | 4.4 ± 0.2 | 4.8 ± 0.3 |
| SO$_2$ pad + Bot | 6.8 ± 0.5b | 1.8 ± 0.2a | 0.6 ± 0.6b | 11.9 ± 4.3a | 4.2 ± 0.5 | 4.8 ± 0.2 |
| F value | 10.1 * | 3.8 * | 6.3 * | 6.4 * | 2.0 [NS] | 0.3 [NS] |

Means within columns followed by the same letters are not statistically different by Fisher's protected LSD test ($p \leq 0.05$). Non-significant ([NS]), *: significant at 5% level of significance.

At seven days at 22 ± 2 °C, significant differences were observed among treatments and the control treatment showed the lowest mass loss in both seasons (5.8% and 1.2% for summer and off-season crops, respectively). In the summer season, a higher mass loss (7.6%) was observed in control with grapes inoculated with *B. cinerea* and in the case of the off-season, a higher mass loss (1.8%) was recorded in SO$_2$-generating pads with grapes inoculated with *B. cinerea* suspension (Table 1).

　　No significant differences were found in terms of shattered berries for both seasons, and at 50 days of cold storage, the means ranged from 0.0 to 0.7% and from 0.7 to 3.1% for the summer season and the off-season, respectively (Table 1). However, statistically differences were found in both seasons at seven days at 22 ± 2 °C, where the control with grapes inoculated with *B. cinerea* suspension showed higher shattered berries (1.5%). On the other hand, in the off-season crop, when grapes were inoculated with *B. cinerea* suspension, combined or not with $SO_2$-generating pads, a higher percentage of shattered berries was found (~12%). However, the percentage of shattered berries was high at seven days at 22 ± 2 °C, especially for the off-season crop.

　　Additionally, there was no change in the berry color index among treatments in both evaluated seasons (Table 1). In the summer season, the berry color ranged from 3.8 to 4.6 (red), while in the off-season crop, the means ranged from 5.1 to 5.5 (red-violet) [30]. During the off-season crop, the anthocyanin accumulation develops under a higher diurnal temperature variation in subtropics, which intensifies berry color and explains these variations. Nevertheless, the original color of the 'BRS Isis' seedless grape was well-preserved in both storage periods.

### 3.3. Chemical Characteristics of Grapes

　　Regarding berry SS content, even though differences among treatments have only been observed at 50 days of cold storage for the summer season, the observed means are in an acceptable range (~14 °Brix), which is a standard for local and international markets of some table grape cultivars [31]. There was no difference in terms of TA and SS/TA of berries among treatments in both evaluated seasons (Table 2).

**Table 2.** Soluble solids (SS), titratable acidity (TA), and SS/TA ratio of 'BRS Isis' seedless table grapes at 50 days of cold storage and at seven days at 22 ± 2 °C during summer and off-season crops.

| | SS (°Brix) | | TA (%) | | SS/TA | |
|---|---|---|---|---|---|---|
| **Treatments** | **At 50 days of cold storage** | | | | | |
| | **Summer season** | **Off-season** | **Summer season** | **Off-season** | **Summer season** | **Off-season** |
| Control | 14.5 ± 0.4a | 14.3 ± 0.2 | 0.6 ± 0.1 | 0.9 ± 0.1 | 23.1 ± 3.4 | 16.8 ± 1.2 |
| $SO_2$ pad | 13.9 ± 0.5b | 14.2 ± 0.5 | 0.6 ± 0.03 | 0.8 ± 0.1 | 23.2 ± 1.4 | 17.7 ± 1.0 |
| Control + Bot | 13.6 ± 0.4b | 14.1 ± 0.2 | 0.6 ± 0.1 | 0.9 ± 0.1 | 23.5 ± 2.0 | 16.6 ± 1.7 |
| $SO_2$ pad + Bot | 14.0 ± 0.3ab | 14.1 ± 0.4 | 0.6 ± 0.03 | 0.8 ± 0.03 | 23.3 ± 0.9 | 18.3 ± 0.6 |
| F value | 3.8 * | 0.3 NS | 0.8 NS | 1.8 NS | 0.04 NS | 2.1 NS |
| | **At seven days at 22 ± 2 °C** | | | | | |
| | **Summer season** | **Off-season** | **Summer season** | **Off-season** | **Summer season** | **Off-season** |
| Control | 15.0 ± 0.4 | 13.9 ± 0.2 | 0.7 ± 0.1 | 0.9 ± 0.1 | 22.9 ± 1.9 | 15.9 ± 1.9 |
| $SO_2$ pad | 14.4 ± 0.6 | 14.1 ± 0.5 | 0.7 ± 0.01 | 0.7 ± 0.03 | 22.1 ± 1.2 | 19.1 ± 0.6 |
| Control + Bot | 14.3 ± 0.5 | 13.8 ± 0.4 | 0.6 ± 0.03 | 0.7 ± 0.03 | 22.2 ± 1.3 | 19.0 ± 1.1 |
| $SO_2$ pad + Bot | 14.6 ± 0.4 | 14.0 ± 1.1 | 0.7 ± 0.03 | 0.8 ± 0.1 | 21.8 ± 0.6 | 18.8 ± 3.4 |
| F value | 2.1 NS | 0.1 NS | 1.1 NS | 2.6 NS | 0.7 NS | 2.8 NS |

Means within columns followed by the same letters are not statistically different by Fisher's protected LSD test ($p \leq 0.05$). Non-significant (NS), *: significant at 5% level of significance.

## 4. Discussion

　　*Botrytis cinerea* Pers. ex Fr., is the most crucial postharvest pathogen attacking table grapes worldwide, causing severe losses of the crop after harvest. It was also observed earlier that the incidence of gray mold was higher, especially in the off-season crop (~30%) when grapes were inoculated with *B. cinerea* suspension. This could be explained by the occurrence of some invisible minor cracks or spots on berry skin caused by powdery mildew *Uncinula necator* (Schwein). Burrill,

which is more prevalent in this season [32]. In the case of the seven day period at 22 ± 2 °C, the incidence was also found high (~50%) in comparison to the 50 day period of cold storage. These results are related with the fact that when grapes are subjected to RT, the disease incidence increases because of the more favorable environmental conditions for fungi development, especially due to the higher air temperature.

Our findings confirm that $SO_2$-generating pads gave better results in controlling *B. cinerea* mold of 'BRS Isis' seedless grapes at 50 days of cold storage and at seven days at 22 ± 2 °C. This new hybrid seedless cultivar showed to be non-sensitive to the amount of $SO_2$ gas released by the evaluated pads, since, at high concentrations, this compound can cause bleaching or premature stem browning and may also damage the fruits, resulting in unwanted conditions. Considering these aspects, combined with cold storage, the $SO_2$-generating pads can be used as a tool to control the gray mold of 'BRS Isis' seedless grapes, at least for a period of 50 days. A similar performance has also been observed in rthe maximum reduction of the disease incidence of 'BRS Vitoria' and 'Italia' table grapes at 50 days of cold storage and at seven days at 22 ± 2 °C, respectively [17,22].

The mass loss is also concerned as it is one of the key factors that determine excellence and quality of table grapes; the more water lost from the produce, the more it develops quality deterioration problems. During the experiment, it was observed that, in the off-season crops, both *B. cinerea* inoculated treatments (the control and $SO_2$-generating pads) showed higher mass loss as compared to non-inoculated treatments. This behavior can be related to the fact that the incidence of gray mold was higher in the off-season crop, which may have caused higher mass loss in inoculated treatments.

Even though during the cold storage period, temperature and relative humidity are controlled to reduce mass loss and extend the shelf-life of table grapes, sometimes mass loss can vary depending on different aspects, i.e., the grape cultivar, harvesting conditions, storage period, and packing materials used. Among other factors, fungal infection, absence, delay in pre-cooling, or high temperatures with low humidity are also the main causes of mass loss [33,34]. For most fresh produce, mass loss percentage should be low to not affect quality attributes (wilting or wrinkling), and the same behavior was observed in the current study of 'BRS Isis' table grapes. The $SO_2$-generating pads were also found to reduce mass loss in both evaluated situations, i.e., grapes inoculated or not with *B. cinerea* suspension. When grapes are subjected to RT, water loss and percentage of shattered berries increases because of the favorable environmental conditions, especially the higher air temperature that reduces the fruit quality, resulting in negatively affecting the grape bunch quality [35].

Regarding chemical characteristics as grape ripening develops under different weather conditions in summer and off-season crops, a slight change between them usually occurs in terms of the main berry chemical properties (Table 2) but does not decrease the grape quality. During cold storage, the recommended temperature for grapes is around 0 °C because most of the variables, such as SS, TA, and SS/TA, remain stable in different grape cultivars at this temperature with controlled atmosphere [36,37].

Table grapes intended for long periods of storage are kept in cold chambers, but each cultivar has a different behavior, i.e., each one has a different storage performance that may comprise from a few days to few weeks, which is determined by its susceptibility to quality defects under low temperatures. Regarding the behavior of the new hybrid seedless grape, 'BRS Isis', this cultivar showed to have a high potential to be stored for long periods under cold chambers, since, after 50 days under these conditions, the bunches packaged with $SO_2$-generating pads and liners were virtually intact. Additionally, the natural incidence of gray mold was found to be very low, which indicates that the natural incidence of *B. cinerea* in this hybrid grape, unlike in some *Vitis vinifera* table grape cultivars, is not a major concern. Even with the use of $SO_2$-generation pads, unwanted situations like bleaching, hairline cracking, and berry softening were not found on the surface of the berries, as these were the main barriers for grape post-harvest quality and maintenance. Shattered berries were also noticed in low levels, which contributes to a better storability and marketability of this grape cultivar in markets.

The period from harvest to marketing of the table grape has a significant importance regarding the maintenance of fruit quality. The results obtained herein showed that the 'BRS Isis' seedless grape

has a large potential for domestic, long distance, and international markets, because a high quality of bunches can be achieved under cold storage for at least 50 days. For domestic markets, including when a long distance transportation is required, the 'BRS Isis' grapes, after being properly packaged, can be transported in refrigerated trucks and easily kept in cold chambers of the market chains and gradually exposed to the consumers with a minimum loss quality. The same could also be applied when the intention is to export this grape overseas, to the European Community or even to North American countries. As long as the cold chain is retained, and considering that it takes up to three weeks to transport a refrigerated container by ship from South America to these regions, 'BRS Isis' seems to fit well for this type of international trade due its high capacity of storage in cold chambers during long periods of up to 50 days or longer. However, since a large proportion of table grapes can be traded overseas, more attention is required for better shipment and quality management.

Finally, for long-term storage and transportation, packaging materials, such as $SO_2$-generating pads and proper liners, also play a crucial role to preserve 'BRS Isis' grapes under cold storage, reducing some unwanted situations and providing a higher efficiency of the $SO_2$ gas for controlling gray mold.

## 5. Conclusions

The new hybrid 'BRS Isis' seedless grape, packaged with $SO_2$-generating pads and plastic liners, has a high potential to be preserved for long periods under cold storage at $1 \pm 1\,^{\circ}$C, at least for 50 days, keeping a very low natural incidence of gray mold, mass loss, and shattered berries.

**Author Contributions:** S.R.R. conceived the research idea. S.A., K.Y., R.C.C., M.S., O.J.C.J., C.H.S., and R.T.d.S. helped to collect the data. S.A. and K.Y. also analyzed the data and wrote the paper.

**Funding:** The authors are grateful for the financial support provided by the Brazilian National Council for Scientific and Technological Development (CNPq) and The World Academy of Sciences (TWAS) for financial support (Grant #190015/2014-4).

**Acknowledgments:** The author gives heartfelt thanks to Antonio Peres for providing grapes and other materials.

**Conflicts of Interest:** The authors declare no conflict of interest.

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
