# Peer review of "Postharvest Preservation of the New Hybrid Seedless Grape, ‘BRS Isis’, Grown Under the Double-Cropping a Year System in a Subtropical Area"

_agronomy, doi:10.3390/agronomy9100603_

Round 1

Reviewer 1 Report

This ms is essentially a study on the postharvest storage life of a new fresh market grape cultivar.  My overall concern is that 2 year old vines were used with 2 harvests of summer and winter.  Despite the assumption that the field is prone to gray mold, these are young vines and one year of harvest may or may not accurately affect gray mold response of the cultivar.  I would hesitate to conclude that this grape would therefore be ready for immediate use in long term storage.

Information regarding the SO2 pads is useful as this type of information is not always readily accessible.  For the experiments, details on the number of replicates used (other than in the abstract), and transformation of data must be done (specifically for the incidence of gray mold and shattered berries).  A 1.5% vs 0.5% difference in shattered berries may or may not be statistically different but is doubtful of biological significance.  Standard deviations would also be useful to add to gauge the amount of noise between treatments and also with color.  And with color, photos indicate that grapes range from green to darker red.  Were grapes specifically selected to represent the dominant grape color or did they range among the colors?

I am assuming the control was no addition of SO2 or other pretreatment? There is little mold, which supports the possibility that this cultivar may be excellent in botrytis resistance, but again, several seasons are needed to conclude this.

Methods on titration-what is semi automated?  Does this mean that a titrimeter was used (model, source) or that a modified system using a digital buret was used.  

There are a few places where English could be improved, such as line 270.

Also, a reference for the statement that botrytis is the primary cause of postharvest loss in grapes is needed (line 51).

Author Response

Reviewer 1

Comments and Suggestions for Authors

Point 1: This ms is essentially a study on the postharvest storage life of a new fresh market grape cultivar.  My overall concern is that 2 year old vines were used with 2 harvests of summer and winter.  Despite the assumption that the field is prone to gray mold, these are young vines and one year of harvest may or may not accurately affect gray mold response of the cultivar.  I would hesitate to conclude that this grape would therefore be ready for immediate use in long term storage.

Response 1: Considering this ‘BRS Isis’ seedless table grape released recently and still a new in the field and the two year old vines mentioned in material and methods was after grafting. In this case, we have four harvesting seasons prior to that one. In Parana state (Brazil), the obtainment of two crops of table grape per year (regular and out of season crops) is possible due the mild winter and the use of budburst stimulators. The first regular crop starts from the end of grapevine dormancy in late winter, and the harvest occurs during summer. Right after that, a new cycle is forced, the grapevines are pruned and forced to sprout again by using budburst stimulators to produce the second crop, and an out of season crop is obtained during autumn.

Point 2: Information regarding the SO2 pads is useful as this type of information is not always readily accessible.

Response 2: In section 2.2 details about SO2-generating pad were included as follow: The slow release SO2-generating pad used in treatments (ii) and (iv) (Osku Hellas®, Grapeguard, Santiago, Chile) contain 73.5% of the active ingredient (Na2S2O5), with 26 cm × 36 cm of dimensions.

Point 3: For the experiments, details on the number of replicates used (other than in the abstract), and transformation of data must be done (specifically for the incidence of gray mold and shattered berries).  A 1.5% vs 0.5% difference in shattered berries may or may not be statistically different but is doubtful of biological significance. 

Response 3: Details regarding the number of replicates were mentioned in section 2.2 details as follow: The completely randomized experimental design was used as statistical model with four treatments and five replications, and each plot consisted of one carton box [each measuring 23 cm × 16 cm × 9 cm (4-kg capacity)]. Concerning the transformation of data, we mentioned in section 2.5 that percentage data were arcsine transformed to normalize variance.

Point 4: Standard deviations would also be useful to add to gauge the amount of noise between treatments and also with color. And with color, photos indicate that grapes range from green to darker red.  Were grapes specifically selected to represent the dominant grape color or did they range among the colors?

Response 4: We agree with the reviewer comment. The standard deviations were added to all tables and figures. Regarding the color, color index was calculated according to CIRG = (180-)/(L*+C*) and details were given in material and methods section. Almost ten readings per each replicate were taken to evaluate color index. The idea here was only to investigate if SO2 has an impact on berries color or not. Finally, the grape berries color ranged among the color average when compared to check treatments. 

Point 5: I am assuming the control was no addition of SO2 or other pretreatment? There is little mold, which supports the possibility that this cultivar may be excellent in botrytis resistance, but again, several seasons are needed to conclude this.

Response 5: We are totally agree with the reviewer to conclude such phenomenon several experiments are needed under various filed conditions. Regarding the control with SO2, in fact, when good agricultural practices are adopted, low natural incidence is expected. In addition, the aim to add SO2 in control was to stimulate the actual practices in the region where the study was carried out.

Point 6: Methods on titration-what is semi automated?  Does this mean that a titrimeter was used (model, source) or that a modified system using a digital buret was used.  

Response 6: The reviewer had right, the tern 'semi automated' is not appropriate here because no digital buret was used. We replaced the term using 'dropwise titration'

Point 7: There are a few places where English could be improved, such as line 270.

Response 7: Following the reviewer suggestion, the whole manuscript was comprehensively revised on the term of English.

Point 8: Also, a reference for the statement that botrytis is the primary cause of postharvest loss in grapes is needed (line 51).

Response 8: As recommended by the reviewer, a reference was added in the text and in the reference list as follow:

Youssef, K., de Oliveira, A. G., Tischer, C.A., Hussain, I., Roberto, S.R., 2019. Synergistic effect of a novel chitosan/silica nanocomposites-based formulation against gray mold of table grapes and its possible mode of action. Int. J. Biol. Macromol. 141, 247–258.

Hashim, A.F., Youssef, K., Abd-Elsalam, K.A., 2019. Ecofriendly nanomaterials for controlling gray mold of table grapes and maintaining postharvest quality. Eur. J. Plant Pathol. 154, 377-388. 

Reviewer 2 Report

Authors present new data on Postharvest performance of the new hybrid seedless grape ‘BRS Isis’,please find below my comments:

line 20: please use italic "Botrytis"

line 55: remove the first "," in [,7,8,9]

line 65: please change "gray mod" with "gray mold"

line 66: please change [13.14.15] with [13,14,15]

line 182 and 183: please use italic "Botrytis"

Please correct "SO2" with "SO2" in figures 1 and 3

Figures and Tables did not report and SE or SD please add it

line 268: please use italic "Botrytis"

I suggest also an English mother-tongue revision

Author Response

Reviewer 2

Comments and Suggestions for Authors

Authors present new data on Postharvest performance of the new hybrid seedless grape ‘BRS Isis’,please find below my comments:

Point 1: line 20: please use italic "Botrytis"

Response 1: Done

Point 2: line 55: remove the first "," in [,7,8,9]

Response 2: Done

Point 8: line 65: please change "gray mod" with "gray mold"

Response 3: Done

Point 8: line 66: please change [13.14.15] with [13,14,15]

Response 3: Done

Point 4: line 182 and 183: please use italic "Botrytis"

Response 4: Done

Point 5: Please correct "SO2" with "SO2" in figures 1 and 3

Response 5: Done

Point 6: Figures and Tables did not report and SE or SD please add it

Response 6: We agree with the reviewer. The standard deviations were added to all tables and figures. 

Point 7: line 268: please use italic "Botrytis"

Response 7: Done

Point 8: I suggest also an English mother-tongue revision

Response 8: Following the reviewer suggestion, the whole manuscript was comprehensively revised on the term of English.
